# NAS-Bench-ASR: Reproducible Neural Architecture Search for Speech Recognition

**Abhinav Mehrotra**[1†], **Alberto Gil C. P. Ramos**[1†], **Sourav Bhattacharya**[1†], **Łukasz Dudziak**[1†],

**Ravichander Vipperla**[1], **Thomas Chau**[1], **Samin Ishtiaq**[1], **Mohamed S. Abdelfattah**[1],

and **Nicholas D. Lane**[1,2]

[1]Samsung AI Center, Cambridge · [2]University of Cambridge · [†]Equal contribution

{a.mehrotra1,a.gilramos,sourav.b1,l.dudziak}@samsung.com

## Abstract

Powered by innovations in novel architecture design, noise tolerance techniques and increasing model capacity, Automatic Speech Recognition (ASR) has made giant strides in reducing word-error-rate over the past decade. ASR models are often trained with *tens of thousand* hours of high quality speech data to produce state-of-the-art (SOTA) results. Industry-scale ASR model training thus remains computationally heavy and time-consuming, and consequently has attracted little attention in adopting automatic techniques. On the other hand, Neural Architecture Search (NAS) has gained a lot of interest in the recent years thanks to its successes in discovering efficient architectures, often outperforming handcrafted alternatives. However, by changing the standard training process into a bi-level optimisation problem, NAS approaches often require significantly more time and computational power compared to single-model training, and at the same time increase complexity of the overall process. As a result, NAS has been predominately applied to problems which do not require as extensive training as ASR, and even then reproducibility of NAS algorithms is often problematic. Lately, a number of benchmark datasets has been introduced to address reproducibility issues by providing NAS researchers with information about performance of different models obtained through exhaustive evaluation. However, these datasets focus mainly on computer vision and NLP tasks and thus suffer from limited coverage of application domains. In order to increase diversity in the existing NAS benchmarks, and at the same time provide systematic study of the effects of architectural choices for ASR, we release NAS-Bench-ASR – the first NAS benchmark for ASR models. The dataset consists of $8,242$ unique models trained on the TIMIT audio dataset for *three* different target epochs, and each starting from *three* different initializations. The dataset also includes runtime measurements of all the models on a diverse set of hardware platforms. Lastly, we show that identified good cell structures in our search space for TIMIT transfer well to a much larger LibriSpeech dataset.

## 1 Introduction

Innovations in Deep Neural Network (DNN) architecture design, data augmentation techniques and a continuous increase in the amount of available high quality training datasets, resulted in a massive reduction in ASR word-error-rate over the past decade [Amodei et al., 2016; Kim et al., 2019; Park et al., 2019; Synnaeve et al., 2020]. However, training ASR models to achieve state-of-the-art performance remains challenging as it requires computationally heavy training process, e.g., often thousands of GPU-hours are needed for good convergence [Amodei et al., 2016; Kahn et al., 2020]. Furthermore, the requirement of hyper-parameter optimizations increases the computational loads in ASR training. Despite the system-level complexities in the training procedure, the importance of novel architecture design has proven extremely important in a variety of application domains including ASR [Chiu & Raffel, 2018; Pratap et al., 2020], computer vision [He et al., 2016; Krizhevsky et al., 2012], and natural-language processing (NLP) [Devlin et al., 2019; Vaswani et al., 2017].

However, architecture design is a non-trivial task and often depends on years of experience, domain knowledge of the researchers and is driven by empirical successes.

Over the past few years, the deep learning community is witnessing a trend in adopting automatic techniques to find neural network architectures over more traditional hand-designed alternatives. NAS algorithms are highly successful in discovering state-of-the-art architectures in various computer vision tasks [Cai et al., 2020; Howard et al., 2019; Lee et al., 2020; Real et al., 2018; Tan et al., 2019; Tan & Le, 2019]. However, many of them suffer from high computational demands, requiring a large number of architecture variations to be trained [Zoph & Le, 2017]. Furthermore, NAS algorithms are often difficult to reproduce by different researchers, mainly due to a non-standard use of training settings, e.g., hyperparameters, and subtle variations in the architecture search spaces [Li & Talwalkar, 2019; Sciuto et al., 2020]. Recently, a number of attempts have been made to mitigate these problems by releasing various *benchmark* datasets for the NAS research community [Dong & Yang, 2020; Klyuchnikov et al., 2020; Siems et al., 2020; Ying et al., 2019]. These datasets usually provide a direct mapping between an architecture variant and its post training performances, which can be used efficiently by a NAS algorithm speeding up the search process and, at the same time, providing common, fully reproducible environment for assessment and comparison of different algorithms. Initial attempts of creating benchmark datasets predominantly focus on image classification tasks (with only one existing work targeting NLP at the time of this writing), and thus suffer from poor application coverage.

We address the lack of coverage problem by introducing a new NAS-benchmak dataset in the domain of ASR, to our best knowledge the very first of its kind. To build the dataset, we have trained $8,242$ unique convolutional neural network architectures on the TIMIT dataset [Garofolo et al., 1993]. We consider convolutional architectures due to their recent successes in the domain of ASR [Pratap et al., 2020; Hannun et al., 2019]. Moreover, convolution based architectures are computationally efficient to run on mobile devices, thus favouring real-time on-device deployment. Our dataset contains multiple runs of the entire training procedure of an architecture, spanning *three* initializations of the network parameters and *three* target epochs, amounting a total of $74,178 = 8,242 \times 3 \times 3$ training runs. In addition to the per epoch validation and final test metrics, such as *Phoneme Error Rate* (PER), and CTC loss, we also provide run-times of the architectures on desktop and embedded GPUs for varying batch size. Furthermore, we compare a number of NAS algorithms [Zoph & Le, 2017; Real et al., 2018; Dudziak et al., 2020; Li et al., 2017; Li & Talwalkar, 2019] on our search space, highlighting potential challenges and differences compared to their performances on existing NAS benchmark dataset. Lastly, we show the transferability of the top architecture cells found on TIMIT to a much larger Librispeech dataset [Panayotov et al., 2015].

In summary, the contributions of this paper are:

- **Design of ASR NAS Search Space.** ASR NAS-Bench is a *first-of-its-kind* search space for convolutional speech models. It facilitates the reproducible study of ASR through NAS methods and thus fills an important gap in the literature. The associated dataset consists of $8,242$ unique *cells* and contains validation and test metrics along with model parameters, FLOPs and on-device run-times[1].

- **Enabling NAS for Large-scale ASR.** Prohibitive training times for non-toy ASR datasets, has prevented NAS from strongly influencing the evolution of ASR architecture design. We show that ASR NAS-Bench is able to support the discovery of cell structures that generalize even to large-scale datasets like Librispeech – *a key breakthrough*. We believe the methodological decisions in this paper will act as a blueprint for future work, where NAS plays a prominent role in ASR design.

- **Validating Existing NAS Algorithm Design.** Existing understanding of NAS is grossly influenced by image-based tasks. By systematically benchmarking popular NAS algorithms, under a rich ASR search space, our findings provide *otherwise lacking scientific support* for prior results.

## 2 RELATED WORK

**NAS Benchmarks.** Ying et al. [2019] inroduced NAS-Bench-101 dataset in an attempt to address the difficulties in reproducing NAS research. It contains over 400K unique image classification

---

[1]The NAS-Bench-ASR dataset and the code can be downloaded from `https://github.com/AbhinavMehrotra/nb-asr`.

models trained on the CIFAR10 dataset. Despite being the biggest NAS dataset yet, not all NAS algorithms can utilize the dataset due to the restrictions it imposes on the maximum number of edges to limit the search space size. To mitigate the limitations, NAS-Bench-201 was introduced, which contains 15K image classification models and includes more diagnostic data [Dong & Yang, 2020]. Concurrently to NAS-Bench-201, NAS-Bench-1shot1 [Zela et al., 2020] and NAS-Bench-301 [Siems et al., 2020] were introduced. NAS-Bench-1shot1 focuses on benchmarking NAS with weight sharing, whereas, NAS-Bench-301 points out the need for surrogate functions for scaling and uses the DARTS search space [Liu et al., 2019] with approximately $10^{18}$ models. Lastly, NAS-Bench-NLP [Klyuchnikov et al., 2020] contains models with custom recurrent cells, which are used to replace traditional layers like LSTMs. The dataset contains around 14K different architectures trained on a language modeling task.

**NAS in Audio Modeling.** Recently, there has been an increasing interest in applying NAS in speech-related tasks, such as *keyword spotting* [Mo et al., 2020; Mazzawi et al., 2019], *speaker verification* [Ding et al., 2020; Qu et al., 2020], and *acoustic scene classification* [Li et al., 2020]. NAS has also been applied in ASR [He et al., 2020; Chen et al., 2020; Kim et al., 2020; Baruwa et al., 2019]. For example, Chen et al. [2020] used a search methodology based on the vanilla DARTS to optimize a CNN-based feature extractor, followed by a fixed Bi-LSTM module and multiple output heads. The authors evaluated the NAS-discovered models under mono- and multi-lingual settings, using the Full Language Pack from IARPA BABEL [Gales et al., 2014], showing improvements over a VGG-based extractor. In contrast, Kim et al. [2020] considered evolution-based search to optimize a micro-cell used within a transformer architecture. The evaluation was done using English and Korean dataset of approximately 100 hours of speech each, under monolingual setting. Similarly, He et al. [2020] used differentiable search, using P-DARTS [Chen et al., 2019] as the base, to optimize a convolutional model without the recurrent *tail*. Unlike the previous approaches, the evaluation is done on Mandarin, while using various training datasets spanning between 170 and 10K hours of speech. Closest to our work is the work by Baruwa et al. [2019], where the authors consider both TIMIT and LibriSpeech datasets, but studied them independently rather than using TIMIT as a proxy for LibriSpeech. Further differences include variations in search space design, e.g., the authors only considered operations arranged in a fixed feed-forward manner, and allowed interleaving convolutions with recurrent blocks. Lastly, current work on NAS for ASR focus mainly on using the search to improve prediction accuracy, however, they often lack solid foundations needed for analysing and reasoning about the overall search process and identifying its limitations. This work presents a large-scale study on the effects of architectural changes on ASR models.

## 3 ASR NAS-Bench

The main purpose of the ASR NAS-Bench dataset is to provide a direct mapping from an architecture instance in the search-space (§3.1) to its training time and final performance metrics. The mapping is designed for any NAS algorithm to quickly navigate the architecture space without requiring the time-consuming and computationally-heavy training procedure. For architecture search we use the TIMIT dataset [Garofolo et al., 1993] and conduct a pilot experiment to select suitable *hyperparameters* used to train $8,242$ unique models.

### 3.1 ASR Architecture Search Space

In-line with existing work [Liu et al., 2019; Pham et al., 2018] and NAS Benchmarks [Ying et al., 2019; Dong & Yang, 2020], we restrict our search to a small feed-forward neural network topology-space, commonly known as *cells*. We repeat and arrange a chosen cell to construct a predefined macro-architecture, which is then trained on the TIMIT dataset.

**Micro-Architecture.** A micro-architecture or a *cell*, as shown in Figure 1(a), is represented by a *directed acyclic graph* (DAG). We consider DAGs with four nodes $T_1, \ldots, T_4$ with corresponding incident tensors $t_1, \ldots, t_4$ and allow two types of edges: *main* and *skip connection* edges. A main edge connects two successive nodes $T_{i-1}$ and $T_i$ in the graph as shown by the solid line-arrows in Figure 1(a). A skip connection edge on the other hand, can connect any two nodes $T_j$ and $T_i$, with the constraint $j < i$ and are depicted as dotted line-arrows in Figure 1(a). Each edge $e_{j \to i}$ represents an operation on the tensor $t_j$. The tensor $t_i$ is computed by summing the results of operations done by all incoming edges on $T_i$ (i.e., $e_{j \to i}$, where $j < i$).

We consider a choice of six operations for the main edges: linear operation, four convolution operations distinguished by choices of $(\mathrm{kernel\_size}, \mathrm{dilation}) \in \{(5, 1), (5, 2), (7, 1), (7, 2)\}$, and a zero operation, which outputs a tensor of zeros with the same shape and type as its input. Skip connection operations on the other hand, can be either the identity operation or the zero operation. We used L2 kernel regularizer with convolution operations and dropouts with linear operations. Within the convolution operations, the size of the kernels (e.g., 5 or 7) is chosen as a trade-off between the audio context duration and the model size. Similarly two dilation factors are considered (e.g., 1 or 2) to investigate the trade-off between audio context duration and time resolution. As the final step of a cell, the value of $t_4$ is further passed through a layer normalization to produce the output of the cell. These design choices are made after we have considered the search space that can be used by the vast majority of the NAS algorithms.

**Macro-Architecture.** One of our focuses in this benchmark is on-device deployment, so the macro-architecture is made up of convolution and unidirectional LSTM blocks as they are computationally ef-

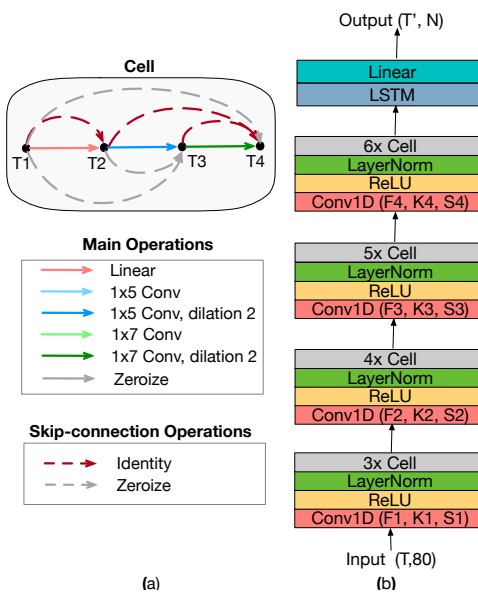

Figure 1: (a) Example of micro-architectures (cells) trained on TIMIT. The architecture space has $8,242$ unique cells. (b) An overview of the macro-architecture.

ficient to run on mobile CPUs. The macro architecture, illustrated in Figure 1(b), is a sequential computation composed of four blocks followed by a unidirectional LSTM and a linear layer. Individual blocks are composed of a convolution layer, an activation layer (ReLU), a layer normalization and a composition of $C_i$ ($i = 1, 2, 3, 4$) search cells, with the same micro architecture across all the blocks. Each block is parametrized by three parameters $F_i$, $K_i$ and $S_i$. These are used to define the number of filters $F_i$, the kernel size $K_i$ and the stride $S_i$ of the convolution layer. Note that $F_i$ is also the number of filters of the convolution layers inside the search cell. We use the following set of parameters to define the macro-architecture while performing all micro-architecture training on the TIMIT dataset: $C_{1:4} = [3, 4, 5, 6]$, $F_{1:4} = [600, 800, 1000, 1200]$, $K_{1:4} = [8, 8, 8, 8]$ and $S_{1:4} = [1, 1, 2, 2]$.

## 3.2 TIMIT DATASET

TIMIT [Garofolo et al., 1993] is one of the earliest datasets designed for evaluating and benchmarking phoneme ASR systems. It is ideally suited for neural architecture search experiments due to its small size and high quality transcriptions. It comprises of $6,300$ utterances from $630$ speakers, amounting to $5.4$ hours of speech. We use the standard training partition of $3,696$ utterances from $462$ speakers. Following Lee & Hon [1989], we split the core test dataset into a test partition, consisting of $24$ speakers, and a validation partition. Inline with kaldi Timit-s5 recipe, the original $61$ phonemes are mapped into a set of $48$ phonemes, which forms the output layer targets for all the models. These 48 phonemes were further folded to a set of 39 during evaluations [Lee & Hon, 1989]. We use 80-dimensional *log-mel* spectrograms computed over $25$ ms sliding window with a stride of $10$ ms as input features. During training, we employ a curriculum learning strategy [Amodei et al., 2016], where we begin training by iterating twice over audio utterances shorter than 1s, then twice over audio utterances shorter than 2s as a warmup phase followed by training on the entire dataset. For efficiency, we also use a batch bucketing strategy, where a batch size of 64 is used for audio utterances smaller than 2s, and a batch size of 32 is used otherwise. We used CTC beam-search decoder with beam-size of 12. Note, we did not use any language model in our experiments in order to avoid further HPO (e.g., weighting factors) optimization. This also helped us to keep the architecture search for Acoustic Model (AM) tractable. All training experiments on TIMIT report the PER on validation and test partitions.

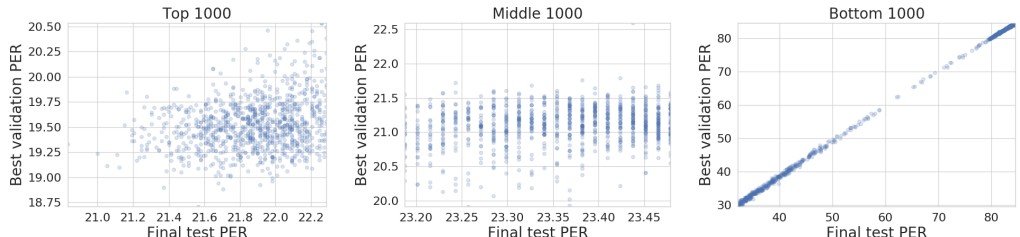

Figure 2: Correlation of best validation and final test PER for top, middle and bottom $1,000$ models.

### 3.3 PILOT EXPERIMENT

As different training procedures can potentially lead to substantially different results [Ying et al., 2019], in this work we employ a single general training procedure for all models. Moreover, we use the same parameter set, e.g., $F, K, S$, learning rate, and decay across all $8,242$ models. In order to select good values of the parameters, we conducted a range of training experiments, where we performed a grid-search to find good macro structure parameters and optimizer settings.

**Macro Structure Parameter Selection.** The macro structure considered in this work has four main blocks (see Figure 1(a)) and we considered the following variations in the pilot study: filters $F \in \{[600, 800, 1000, 1200], [900, 1100, 1300, 1500], [1200, 1300, 1500, 1700]\}$, kernel sizes $K \in \{[6, 8, 10, 12], [8, 8, 8, 8], [10, 10, 10, 10], [12, 12, 12, 12]\}$, and time reduction via strides $S \in \{[2, 2, 2, 1], [1, 1, 2, 2], [2, 2, 1, 1]\}$.

**Optimizer Setting for Training.** We further explored good ranges of the learning rate, decay factor, and start epoch for an exponential decay learning rate scheduler used in conjunction with the Adam optimizer. Specifically, we considered the learning rate (LR) $\in [10^{-3}, 10^{-4}, 5 * 10^{-5}, 10^{-5}]$, the decay factor $\in [0.9, 0.95, 0.99]$, and the LR decay start epoch $\in [5, 10, 15]$.

Finally, we randomly chose *five* micro architectures among all possible $8,242$ cells and for each cell we conducted the grid search over the aforementioned parameters. Specifically, for each cell we created a model from a particular macro architecture, which we then trained by selecting a particular optimizer setting on TIMIT. We identified the parametric setting that resulted in the lowest PER across all *five* cells in the experiments and used the same parameters in all latter model training experiments. The best macro structure parameters are presented above (see §3.1), whereas the best LR was $10^{-4}$, and the decay factor and start epoch were: (*i*) 0.9 and 5 for target epoch 40, (*ii*) 0.631 and 2 for target epoch 10, and (*iii*) 0.398 and 1 for target epoch 5 respectively. The decay factor and start epoch are chosen such that the learning rate of the optimizer reaches the same value at the end of the target epoch of training.

### 3.4 TRAINING SETUP

Individual models are trained using a TensorFlow-based training pipeline running on a single GPU. We leveraged NVIDIA V100 and P40 GPUs, and decreased training time by increasing throughput via the bucketing strategy based on the audio length. Metrics such as CTC loss and *Phoneme Error Rate* (PER) for TIMIT were used to measure model performances. Specifically, train and validation metrics were logged at each epoch, and the test metrics were logged at the end of training for the best model (refered to as *final test PER*). Our dataset contains logs of each of the $8,242$ models trained with three different seeds and for three target epochs (5, 10 and 40), thus generating a total of $74,178$ model training traces. Additionally, we computed the number of parameters and floating point operations (FLOPs) for each of the architectures and measured their latency on two commonly used hardware platforms: Tesla 1080Ti and Jetson Nano.

## 4 ANALYSIS OF NAS-BENCH-ASR DATASET

As described in §3.1, our cell search space is constructed by selecting 3 main operations from a set of 6 candidates, and then selecting either identity or zeroize operation for the 6 skip connection edges (see Figure 1(a)). This amounts to $13,824$ $(6^3 \times 2^6)$ possible instances for the search-space. How-

Table 1: Summary of test *vs.* validation analysis on NAS-Bench-ASR, including three subgroups. Spearman-correlation between the test and validation performances are also presented.

| Group | Positions | Test PER (%) | | | Val. PER (%) | | | Test-Val Corr. |
|---|---|---|---|---|---|---|---|---|
| | | Min. | Max. | Avg. | Min. | Max. | Avg. | |
| Overall | All | 20.83 | 90.28 | 30.37±36.47 | 18.71 | 90.41 | 28.21±38.03 | 0.8519 |
| Top | 1-1000 | 20.83 | 22.29 | 21.87±0.52 | 18.71 | 20.54 | 19.52±0.51 | 0.2037 |
| Middle | 2000-3000 | 23.19 | 23.48 | 23.36±0.17 | 19.91 | 22.59 | 21.11±0.58 | 0.1408 |
| Bottom | 7000-8000 | 32.16 | 84.53 | 65.03±42.68 | 29.57 | 84.6 | 64.52±44.21 | 0.9977 |

Table 2: Comparison of correlations between test and validation accuracies reported in the existing image classification benchmarks NAS-Bench-101, NB-Bench-201, and our NAS-Bench-ASR.

| Group | NAS-Bench-101 | NAS-Bench-201 | | | NAS-Bench-ASR |
|---|---|---|---|---|---|
| | | CIFAR-10 | CIFAR-100 | ImageNet-16 | |
| Overall | 0.989 | 0.993 | 0.987 | 0.997 | 0.8519 |
| Top 1000 | 0.356 | 0.855 | 0.613 | 0.827 | 0.2037 |
| Top 5% | 0.573 | 0.826 | 0.596 | 0.796 | 0.141 |

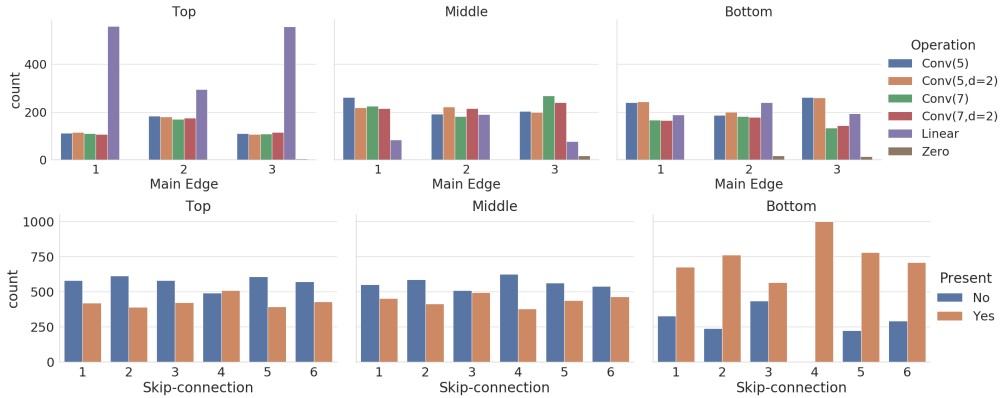

Figure 3: Distribution of selected edges for each node (first row) and distribution of skip-connections for possible edges (second row). Top, middle and bottom are grouped by final test PER.

ever, the use of zero operations introduces computational *graph-isomorphism*. To identify unique architectures, following [Ying et al., 2019; Dudziak et al., 2020], we first identify nodes with zero operations and disconnect them from their neighbors. Then we perform a reachability test and remove all nodes that can not be reached from the input or output nodes of the cell. Finally, we end up with a minimized graph representation, which we hash using an iterative graph invariant approach [Ying, 2019]. After accounting for isomorphism and discarding the empty graph from the list of valid models, we found 8, 242 unique architectures, out of which 8, 000 are without any zero operations.

**Distribution of Model Performances.** Analyzing the model training results we found that the majority of the models have converged and achieved a final test PER within a small range. Specifically, 83.3% (6, 869) of the models fall below a PER of 26%, with the best model reaching test PER of 20.83%. The remaining 16.7% of the models form a rather evenly-distributed tail, achieving PER values between 26% to 100%. Because of the high density of points below 26% PER and overall high range of PER values, we conducted our analysis by considering three subgroups of models: TOP models are the top 1, 000 architectures according to test PER, MIDDLE are models at positions between 2, 000 and 3, 000 in the test PER ranking, and BOTTOM are between 7, 000 and 8000.

**Correlation Between Validation and Test PERs.** Figure 2 shows the correlation between the test PER and the best validation PER. The results indicate that there is a moderate correlation for top models, weak for middle ones, and strong for worst ones. An overall summary is also presented in Table 1. In general, correlation between validation and test accuracy among the top performing models is significantly lower than for image classification benchmarks, which is highlighted in Table 2, can potentially pose additional challenge to NAS algorithms.

**Distribution of Operations.** Figure 3 (first row) presents the distribution of selected operations among main edges across top, middle and bottom most models. We found that the top models have the highest number of *linear* operations on the main edges, while the middle and bottom groups have low linear operations but higher *convolution* operations. Figure 3 (second row) also highlights that the worst models (bottom group) also have the highest selection of *skip-connections*. Figure 4 further confirms this observations showing how significantly worse the models in the search space become as they include less linear layers, or more than 2 skip connections.

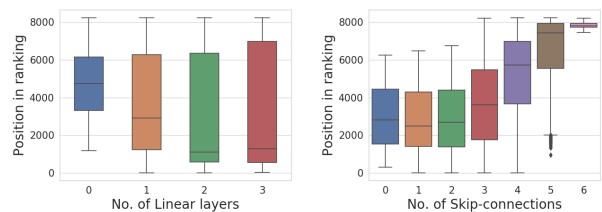

Figure 4: Ranking of models according to their test PER, when they are grouped according to the number of Linear layers (left) and skip-connections (right) used in their designs.

Interestingly, we found that the micro cells in our search space yielding the best validation PER is similar to the structure found in the state-of-the-art convolution models such as time-depth-separable (TDS) models [Pratap et al., 2020]. Figure 5 shows the TDS-cell and the micro-cells with best validation and test PERs (from left to right). This indicates the potential of conducting NAS on smaller datasets to discover new architectures yielding state-of-the-art results on larger datasets.

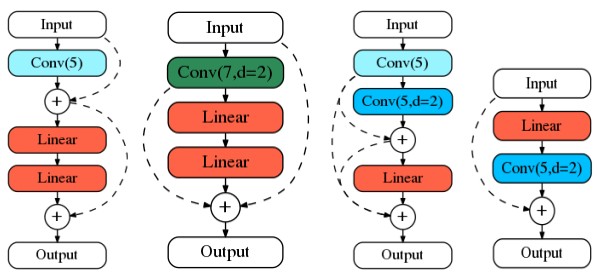

Figure 5: Cell architectures in our search space (from left to right): TDS-like, with best validation and test PERs, and one of the efficient pareto points. Note that the TDS-like cell could also have different convolution selected as the first operation.

**Latency, Params, FLOPs and PER.** Figure 6 investigates the relationship among test PER, number of parameters and latency of models in the dataset. FLOPS counts are closely correlated to the number of parameters so are not shown for simplicity. Although this figure shows only latency measured on a desktop GPU (NVIDIA GeForce GTX 1080 Ti) with batch size of 32, the NAS-Bench-ASR includes latency measured on different devices with varying batch sizes. Also shown in the figure are 13 Pareto-optimal models, indicated by purple (x) markers. Also, the best models on test and validation dataset are highlighted in red markers. TDS-like models ([Pratap et al., 2020]) achieve test PER between 22.7% and 23.45%, and the best validation PER model has test PER of 22.13%. There are Pareto-optimal models that achieve much lower latency without significant increase in PER, for instance, the model indicated by violet (♦) has one-fifth of the latency of the best test PER model and still has a test PER of 21.98%. The distribution of coloured clusters along the y axis suggests that the number of parameters is not a strong factor to determine the accuracy of models. Analogically, models with similar number of parameters can lead to very different latency.

## 5 TRANSFERABILITY

Training an ASR architecture to achieve state-of-the-art WER is a slow process, as it often requires iterating over tens of thousands hours of speech data, and potentially over various hyper-parameter choices. Even when distributed across multiple GPUs, this process can take days or weeks [Amodei et al., 2016; Kahn et al., 2020]. Since the largest strides in lowering WERs have resulted from new architectures being developed manually by speech experts slowly over decades, it would be beneficial to scale such exploration through NAS. However, performing NAS directly on large scale ASR datasets needed for peak performance is not practical due to significantly more computational costs. An issue which is further exacerbated when one is also interested in architectures yielding lower latency and computational cost.

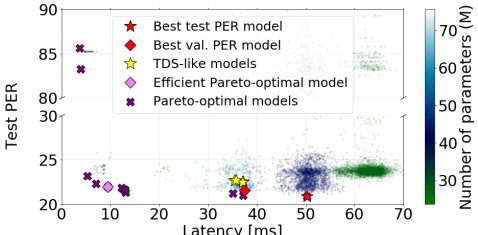 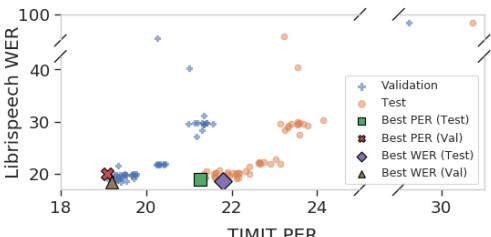

Figure 6: Test PER versus latency, color-coded by the number of parameters. Best models, Pareto-optimal models, and selected TDS-like models are highlighted.

Figure 7: Transfer between TIMIT and Librispeech on validation and test datasets with high validation and test correlation of $0.85$ and $0.86$, respectively.

For this reason, we explore how well new architectures found by NAS on TIMIT (5 hours) transfer to Librispeech (100 hours). Specifically, we investigate the correlation between the PER of best/random/worst performing architectures found on TIMIT, and the *Label Error Rate* (LER) (or WER) obtained when training the same architectures on Librispeech. If a high correlation is present, this would mean that one could lower the computational requirements of NAS by running it on a smaller datasets. In our case this would mean a 20x speedup in NAS, and in general this would open a new avenue for research, namely on how much one could save by running NAS on different smaller datasets, or running NAS on a subset of a larger dataset extracted by current or next generation data summarization techniques.

**Librispeech Dataset.** Librispeech [Panayotov et al., 2015] is a widely used ASR training and benchmarking database of about $1,000$ hours of speech derived from audiobooks. The training dataset is partitioned into clean-100, clean-360 and other-500 sets with the clean partitions exhibiting much higher SNR. The test and dev sets are also partitioned similarly. In our work, we have used clean-100 partition for training and dev-clean and test-clean for validation and testing. The choice of 100 hours partition is to strike a balance between building a decent large scale model, while still keeping the computational cost involved in architecture search experiments plausible. Previous literature indicates that the models trained on 100 hours partition are about 1-1.5% absolute [Panayotov et al., 2015] behind the models trained on $1,000$ hours when evaluated on test-clean set.

**Training.** Librispeech models were trained with $4$ GPUs with Horovod distributed framework. We use 80-dimensional log-mel features computed over 25ms Hamming windows strided by 10ms. Guided by latency considerations, we used a vocabulary of 780 sub-word tokens as output. Model performances are measured by tracking CTC loss, LER, and WER on the validation dataset during training and on test dataset post-training.

**Pilot Experiments.** Similarly to §3.3, we also conducted a pilot experiment to determine appropriate macro structure parameters and hyper-parameters to train the models for Librispeech. To promote transferability between datasets at a reasonable compute cost, with the micro cell of the best model found on TIMIT dataset (out of $8,242$), we conducted another grid search but instead on Librispeech dataset. Accordingly, we chose the final number of filters, kernel size, dilation, learning rate, decay factor and start epoch based on lowest LER/WER on Librispeech.

**Correlation Analysis.** For a meaningful analysis we selected 100 architectures from our search space based on their TIMIT PER, namely we took the best 20, 70 random, and the worst 10. We then plugged the micro cell architectures into the macro architecture for Librispeech. Finally, we trained these models on Librispeech with the hyper-parameters reported earlier and computed their LERs/WERs. We observe that models that didn't converge for TIMIT also didn't converge for Librispeech. For this reason, when looking at the strength of transferability between TIMIT and Librispeech we considered models that had TIMIT PER lower than $50\%$. As shown in Figure 7, there is a high correlation between TIMIT PER and Librispeech WER. In particular, we observe $0.85$ correlation between validation metrics, and $0.86$ for test metrics. Thus, it is possible to do indirect NAS for cell micro architecture on Librispeech by first applying NAS on TIMIT, and then reusing the identified cell structures in a macro architecture suitable for Librispeech.

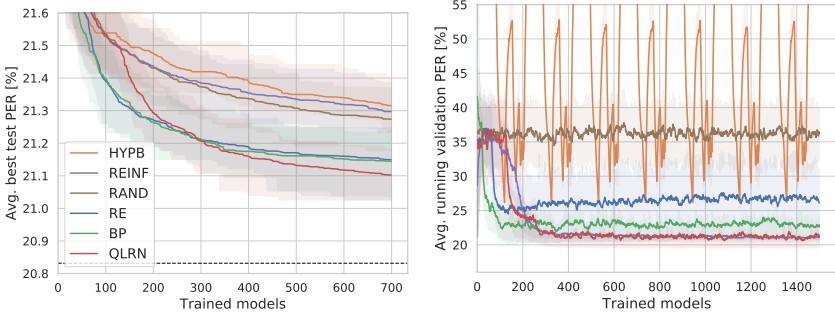

Figure 8: Performance of NAS algorithms on our ASR search space, where shaded regions mark interquartile ranges. The dashed line represents the best model in the dataset across all seeds.

## 6 NAS Experiments

Following existing NAS benchmarks, we include results from running different NAS algorithms on our search space, using the NAS-Bench-ASR to query for accuracy of models. We consider the following standard NAS algorithms: Regularized Evolution (RE) [Real et al., 2018], Hyperband (HYPB) [Li et al., 2017], REINFORCE with an LSTM-based controller (REINF) [Zoph & Le, 2017], Binary Predictor (BP) [Dudziak et al., 2020], and Random Search (RAND) [Li & Talwalkar, 2019]. We also include an algorithm based on Deep Q-Learning (QLRN), which behaves similarly to RE but tries to chain mutations (actions) and learns what mutations can eventually lead to better results. We report results as functions of "trained models" to unify notion of cost. Because in the full training setting we trained models for 40 epochs, if an algorithm trains a model for fewer epochs than that (e.g. HYPB) we consider the cost of such a training to be $n/40$ of a "trained model", where $n$ is the number of epochs used ($0 < n \leq 40$). We run each algorithm 100 times and present results in Figure 8 as average test PER of the best model found (all algorithms were using validation accuracy as their optimization target), and average validation PER of the most recently trained model (after smoothing the curve using exponential moving average with weight 0.9). We found that none of the algorithms were able to consistently find the best model – this is somewhat surprising given the moderate size our search space and the fact that algorithms like RE are able to consistently find the best model on NAS-Bench-201, given similar budget. Another interesting observation is related to the weak correlation between validation and test accuracies mentioned in the previous sections. By comparing the figures on the right and left sides (Figure 8), we can see that algorithms which tend to better optimize their rewards (validation accuracy) as search progresses are not necessarily better in the test PER objective – this is especially visible in the case of REINFORCE, which significantly outperforms random search if validation accuracy is considered, but when evaluated using test PER it presents comparable results. Another surprise is weak performance of Hyperband – the only algorithm that has the ability to adjust number of epochs to train models. From our observations it seems that many of our models converge to relatively good PER values fast (after 5-10 epochs) so we expected Hyperband to be able to leverage this property. However, it seems that in the case of our search space this is not so helpful – potentially because models might still change relative ordering as they are trained for more epochs despite the fact that they have already achieved decent performance.

## 7 Conclusions

We introduced NAS-Bench-ASR, a first comprehensive dataset for allowing computationally inexpensive training/test time evaluation of model performances, while conducting NAS research in the domain of ASR. We trained $8,242$ unique models on the TIMIT dataset for *three* initializations and *three* target epochs. In addition to the common training metrics in ASR, e.g., PER, CTC-loss, we also provide information on the number of parameters, FLOPS and latency of running all the models on two hardware platforms for varying batch sizes. Model run-times are especially beneficial for NAS researchers interested in optimizing architectures for efficient on-device deployment. We presented a comprehensive analysis of our dataset and evaluated the performances of a number of NAS algorithms on it. Lastly, we show that good cell structures as identified on the TIMIT dataset transfer well to Librispeech, which paves the way for affordable NAS on ASR domain.

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

# A APPENDIX

Our dataset comprises of 8,242 unique models that were trained for three target epochs and each starting with three different initializations. In this appendix we present the results for the models with different initializations and target epochs.

## A.1 CORRELATION BETWEEN VALIDATION AND TEST PERS FOR THE RUNS WITH DIFFERENT TARGET EPOCHS

In this section we present the correlation of validation and test PERs between different target epochs. Specifically, we correlate each run of target epoch 40 (i.e., runs starting with unique initialization and trained for 40 epochs) with the runs for target epoch 5 and 10. This analysis informs us if reduced training (i.e., target epoch of 5 or 10) can be used as a good proxy for NAS. As shown in Table 3, there is weak correlation of target epoch 40 and 5. However, there is a moderate correlation between target epoch 40 and 10, suggesting that the reduced training might be useful for performing NAS.

Table 3: Correlation of validation and test PERs between different target epochs.

| E-40 | E-5 | | E-10 | |
|---|---|---|---|---|
| | **Validation** | **Test** | **Validation** | **Test** |
| Run 1 | 0.3349 | 0.3097 | 0.5493 | 0.5029 |
| Run 2 | 0.2986 | 0.3022 | 0.5138 | 0.4793 |
| Run 3 | 0.3234 | 0.3362 | 0.526 | 0.5112 |

## A.2 CORRELATION BETWEEN VALIDATION AND TEST PERS

In this section we present the correlation between the final test PER and the best validation PER for the three runs for target epoch 40 starting with different initializations, which is extension of results presented in Figure 2. As shown in Figure 9, the results for all seeds are similar. Specifically, there is moderate correlation for top models, weak for middle ones, and strong for worst ones.

## A.3 CELL ARCHITECTURES IN OUR SEARCH SPACE

In this section we present the micro-cell architectures in our search space that achieved best validation and test PERs for the three runs starting with different initializations. Figure 10 shows the micro-cell architectures that achieved best validation PERs (in top row) and best test PERs (in bottom row). We observe that the micro-cell architectures of top models often start with two convolution layers followed by a linear layer (which is consistent across all architectures).

## A.4 IMPACT OF DIFFERENT OPERATIONS-COUNT ON MODEL PERFORMANCE

In this section we present the impact of the number of selected operation on the model performance. As shown in Figure 11, models with more convolution layers, or with no linear layer are ranked as worst performing models. At the same time, models with a zero ops in the main edge have poor performance. This indicates that the top performing models are designed such that all nodes comprise of linear and convolution layers. We also observe that an increase in the number of skip connections deteriorates performance significantly, with surprisingly two skip connections yielding similar results to the traditional one skip connection, which aims to learn a perturbation of the identity rather than a generic transformation between input and output nodes.

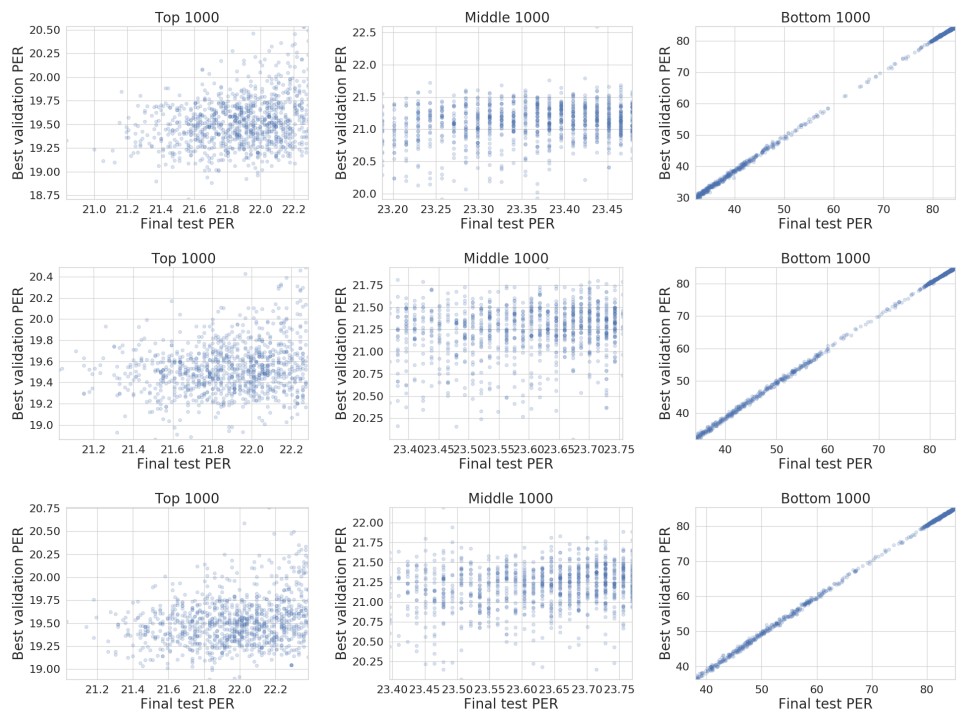

Figure 9: Correlation of best validation and final test PER. Figures in the columns are for top, middle and bottom $1,000$ models. Each row represents the results for runs with unique initialization.

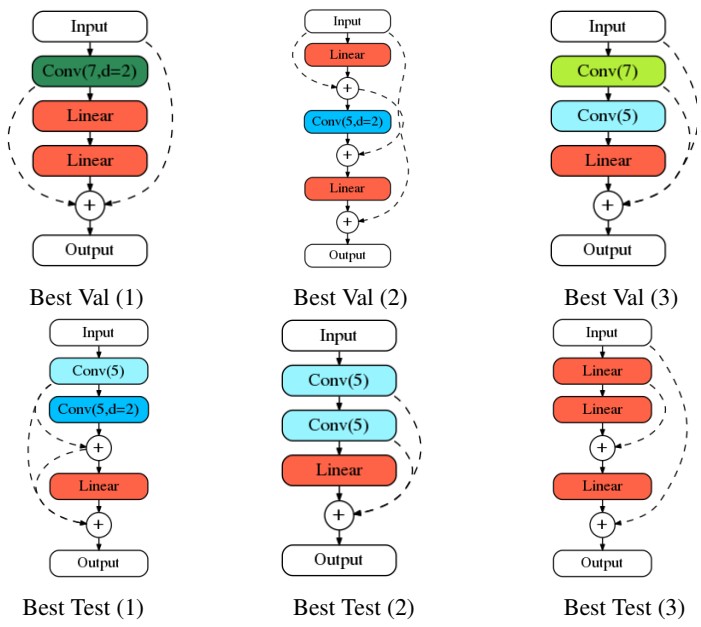

Figure 10: Cell architectures in our search space with best validation and test PERs for the three runs.

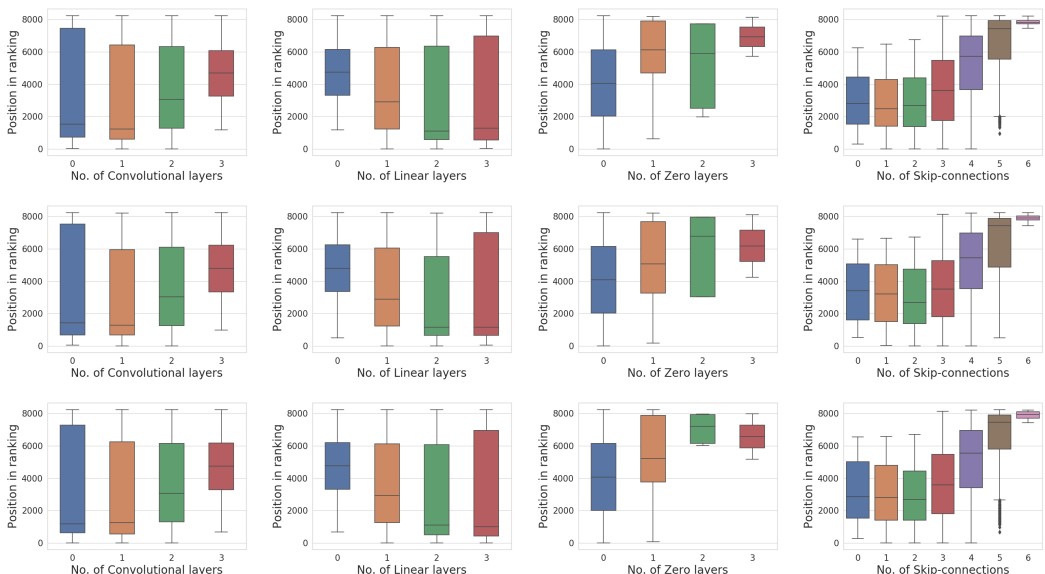

Figure 11: Positions of models in the ranking according to their test PER, when they are grouped according to the number of Convolution layer (first column), Linear layer (second column), Zero op (third column) and Skip-connections (last column). Each row correspond to the results for a run with unique inititialization.

