# OpenReview forum: "NAS-Bench-ASR: Reproducible Neural Architecture Search for Speech Recognition"
_ICLR.cc/2021/Conference — ICLR 2021 Poster_

### Official Review · AnonReviewer2 · 2020-10-23
**results not close to sota**

**Rating:** 4
**Confidence:** 5

**Review:**

The authors contribute to the NAS literature by presenting a framework that works decently well on small ASR tasks, specifically TIMIT. They make judicious decisions regard the macro and micro cells that are then swept over. They also show that there is some correlation between training for TIMIT and tasks that have more data, such as librispeech. The experiments look to have been done carefully.

My chief issue with the work is how far the results are from sota on any of the tasks. Their NAS search for TIMIT only yields PER of 21.93 on test, 19.55 on val. wav2vec 2.0 [1] gets 8.3 test, 7.4 val. Authors may argue the wav2vec results are pre-trained, and I would argue that the authors should also do that. However, [2] gets 13.8% on timit test with training being from scratch. Similarly, their best librispeech wer is 19, which is _very far_ from sota from the same paper. Even their transfer correlations between TIMIT and LibriSpeech are not very high.

It is challenging to evaluate their results when they are so far from sota. Authors should resubmit their paper with updated results.

[1] Baevski, et al. wav2vec 2.0: A Framework for Self-Supervised Learning of Speech Representations. https://arxiv.org/pdf/2006.11477.pdf

[2] Ravanelli, et al. THE PYTORCH-KALDI SPEECH RECOGNITION TOOLKIT. https://arxiv.org/pdf/1811.07453v2.pdf


========================================================================

I thank the authors for their detailed rebuttal. However, their accuracies are still very below sota. Therefore, I am inclined to stick to my original review and rating.

---

> ### Author Response · Authors · 2020-11-22
> **It's not about being close to SOTA (part 2/2)**
>
> (continued from the previous comment)
>
>  - The reviewer claims that it is hard to evaluate our results but we think that’s not true. Of course, were our results close to sota, it would be much easier to judge them, but even in the current shape we think there is plenty of arguments supporting our work:
>       - Firstly, we would like to emphasize that NAS benchmarks are all behind relevant SOTA levels (see the table below) - this is caused by the fact that their goal (just like ours) is not related to beating SOTA but rather to provide insights into the effects of different architectural choices on the final performance, and providing a robust way of accessing searching algorithms easily.
>       - In the context of the above, a NAS benchmark is valuable as long as the insights it provides are relevant for a particular domain(s). In that regard we’d like to point out that:
>            - the best architectures found in our search space match surprisingly well the ones found in manually designed SOTA models - that suggests that the results are meaningful as they do not obviously contradict what is currently known.
>
>            - at the same time the results show that more efficient variations can be found, which is also expected considering the success of NAS in other domains. It is important to clarify that we do not claim that the best model in our search space is ultimately the best to use in any SOTA-oriented research but rather we’d argue that a NAS algorithm which is able to identify the best model in our search space quicker is more likely to be successful when used in the SOTA-oriented research - which is the point of a good NAS benchmark.
>
>           - from a more NAS-oriented point of view, the insights about the number of skip connections on a model’s performance is both interesting and potentially relevant for differentiable search which is known to be sensitive to choices regarding skip-connections (addressed for example in [6], for more recent comment on the problem see for example section 3.2 from one of the concurrent submissions [7]). The fact that our observations are somewhat aligned with this relatively distant problem in NAS proves, in our opinion, that our benchmark is potentially useful and thus can be objectively judged as a good contribution regardless of the gap to SOTA.
>
> We hope that the above response shows clearly that the contributions of our work are both relevant in the context of both NAS and ASR and remain valid even considering the fact that our accuracy values remain behind the current SOTA. Similar to other NAS benchmarks, we expect our work to fuel future ASR work by making it easier to use NAS and thus, directly, contributing to future gains in the SOTA-oriented research.
>
>
> |  | Best Reported Accuracy | Models from literature |
> | -- | -- | -- |
> | NAS-Bench-101 (2019) | 94.32 (CIFAR-10)  | (see below) |
> | NAS-Bench-201 (2020) | 94.37 (CIFAR-10) | 94.6 [8] (2017), 97.92 [9] (2019), 99.3 [10] (2019) |
> | | 73.51 (CIFAR-100) | 82.82 [11] (2017) |
> | |  47.31 (ImageNet-16-120) |  |
> | NAS-Bench-NLP (2020) | 78.5 (PTB, word level) | 78.4 [12] (2014), 31.3 [13] (2019) |
>
> **References:**
>
> [3] Zang et al. Towards End-to-End Speech Recognition with Deep Convolutional Neural Networks. 2017.
>
> [4] L. Dudziak et al. “BRP-NAS: Prediction-based NAS using GCNs.” NeurIPS (2020)
>
> [5] https://openreview.net/forum?id=0cmMMy8J5q
>
> [6] X. Chen et al. “Progressive Differentiable Architecture Search: Bridging the Depth Gap between Search and Evaluation.” ICCV (2019)
>
> [7] https://openreview.net/forum?id=PKubaeJkw3
>
> [8] E. Real et al. “Large-Scale Evolution of Image Classifiers.” ICML (2017)
>
> [9] H. Cai et al. “ProxylessNAS: Direct Neural Architecture Search on Target Task and Hardware.” ICLR (2019)
>
> [10] L. Wang et al. “Sample-Efficient Neural Architecture Search by Learning Action Space.” arXiv preprint arXiv:1906.06832 (2019)
>
> [11] G. Huang et al. “Densely Connected Convolutional Networks.” CVPR (2017)
>
> [12] W. Zaremba et al. “Recurrent Neural Network Regularization.” arXiv preprint 	arXiv:1409.2329 (2017)
>
> [13] C. Wang et al. “Language Models with Transformers” arXiv preprint arXiv:1904.09408 (2019)

---

> ### Author Response · Authors · 2020-11-23
> **It's not about being close to SOTA (part 1/2)**
>
> We thank the reviewer for his/her valuable comments and present our response below.
>
> Indeed, if evaluated purely in the context of beating/matching SOTA results, the results presented in our work might not look convincing. However, we think that limiting one’s judgment to this criteria alone is not fair as it misses a lot of benefits our work brings, due to a different context (beating SOTA vs. helping NAS/ASR research). To further explain our point of view, we’d like to present a list of observations made in the paper which we think are important, together with justification why we think the gap between SOTA and our results does not impact our contributions.
>
>   - Our models achieve accuracy much closer to the ones presented in the literature for similar models [3] than the numbers presented by the reviewer. Even though we agree that [3] does not represent the current SOTA, we think it is important to mention it because it shows that there is nothing fundamentally wrong with the way we train our models and our results are still representative for the chosen setting. To further tackle the problem of accuracy:
>       - Pre-training the models (as in [1]) is both infeasible, if done for all models, and incorrect as it would likely invalidate our transferability experiments. Answering the question whether training on TIMIT can be successfully used as a proxy for much larger LibriSpeech (~200x larger) tackles one of the core problems in NAS concerning finding cheaper ways of evaluating models - in that context, correlation of 0.87 is high enough in order to consider a methodology successful. For example, one of the SOTA NAS methods achieves correlation of 0.85 when learning the ordering of models in the NB2 search space [4]. Although the same paper also shows that higher correlation does not have to imply better NAS performance, in order for us to make any more specific comments we’d need to dive deep into technicalities of specific searching algorithms (as the degree in which different algorithms are sensitive to correlation can vary) which is way outside the scope of our work. However, we think that this alone serves as a good counter-argument to the claim that correlation of 0.87 is not very high. To provide further details about correlations of proxy tasks and their impact on NAS, we invite the reviewer to take a look at one of the concurrent submissions which happens to talk about this specific problem extensively, thus providing a good “survey” [5].
>       - Compared to [2], our models don’t use attention-based CTC. We admit that including attention-based mechanisms in our search space would potentially make our work more appealing but we decided to ignore it in order to keep the search space size limited. Even though we do not consider attention, we'd argue that our findings are still relevant - see further comments for details why.

---

### Official Review · AnonReviewer4 · 2020-10-28
**A sound paper with somewhat discouraging results**

**Rating:** 6
**Confidence:** 4

**Review:**

This paper studies neural architecture search for automatic speech recognition. The approach is to first search over small, reusable networks, called cells, and then applies the cells to a template network. The cells are learned with phonetic recognition on TIMIT and validated on letter recognition on LibriSpeech.

The approach strikes a good balance between having a large search space and the computation cost of the search. I will discuss a few weaknesses in detail, but these weaknesses won't be known prior to performing the experiments in the paper.

The presentation of the paper is also done well. I have no trouble following the paper from start to finish.

The weakness of the paper is the absolute PERs on TIMIT and WERs LibriSpeech. The best PER achieved in this paper, 21.1% is quite high in today's standard. In (Graves et al., 2013), the numbers are around 18%. The best WERs achieved in Figure 7 are high in the teens. In (Hsu et al., 2020), the number for training on the 100 hours of LibriSpeech is around 14%.

It is unclear if the paper uses any regularizer at all when training the models. Even adding some amount of dropout would help the final numbers.

The discrepancy between the numbers in the paper and others makes me wonder the search over the cells is a wrong direction to begin with. Maybe it is the things held fixed that play the role of achieving the best numbers. For example, the macro architecture is held fixed, the optimizer is held fixed, the learning rate schedules are more or less fixed. The macro architecture might play a critical role here. Typically, the competitive architectures require many layers of LSTMs instead of one used in the paper. It is quite discouraging that the models, discovered by NAS after spending so much compute, are not competitive to baseline models reported in other papers.


Hybrid Speech Recognition with Deep Bidirectional LSTM
Alex Graves, Navdeep Jaitly, and Abdel-rahman Mohamed
ASRU, 2013

Semi-Supervised Speech Recognition via Local Prior Matching
Wei-Ning Hsu, Ann Lee, Gabriel Synnaeve, and Awni Hannun
arXiv:2002.10336

---

> ### Author Response · Authors · 2020-11-23
> **Clarification of missing points in the paper**
>
> We would like to thank the reviewer for his/her time and effort in giving us valuable feedback. In the following we present our responses to his/her comments.
>
> **Re. 1. The best PER achieved in this paper, 21.1% is quite high in today's standard. In (Graves et al., 2013), the numbers are around 18%. The best WERs achieved in Figure 7 are high in the teens. In (Hsu et al., 2020), the number for training on the 100 hours of LibriSpeech is around 14%.**
>
> Please see our reply to Reviewer-2.
>
>
> **Re. 2. It is unclear if the paper uses any regularizer at all when training the models.**
>
> Yes, we used L2 kernel regularizer with convolution layers and dropouts with dense layers. We will incorporate this information in the revised paper.
>
> **Re. 3. The discrepancy between the numbers in the paper and others makes me wonder the search over the cells is a wrong direction to begin with. Maybe it is the things held fixed that play the role of achieving the best numbers. For example, the macro architecture is held fixed, the optimizer is held fixed, the learning rate schedules are more or less fixed. The macro architecture might play a critical role here. Typically, the competitive architectures require many layers of LSTMs instead of one used in the paper. It is quite discouraging that the models, discovered by NAS after spending so much compute, are not competitive to baseline models reported in other papers.**
>
> Before performing a search on micro-cells, we did a search for a good macro architecture, learning rate, and learning rate decay factor. In order to select good values of the parameters, we conducted a range of training experiments to find good macro structure parameters and optimizer settings. Please see Section 3.3 for more details.

---

### Official Review · AnonReviewer3 · 2020-10-30
**Interesting research on NAS in ASR domain**

**Rating:** 6
**Confidence:** 4

**Review:**

Motivation and summary:
A lot of research works have been done for NAS-benchmark in the domain of computer vision (and also NLP). This paper introduces a NAS-benchmark dataset for ASR. The authors release a NAS-Bench dataset which would benefit both ASR model architecture search and reproducible NAS research.
This is an interesting and pioneer work in the ASR domain. The NAS-Bench-ASR is built upon TIMIT, a relatively small corpus for speech phonetic recognition. Besides the careful analysis on the designed NAS dataset, the authors also evaluated a few NAS algorithms, and showed that good cell structures identified on the TIMIT dataset aligned with some existing convolutional ASR models in the literature, and can be transferred to Libirspeech. I have below questions/concerns:

Regarding the design of the macro-Architecture:
Why the design only uses unidirectional, rather than bidirectional LSTM? Is the paper focusing on on-device deployment?

Regarding the performance on TIMIT:
The best performance on TIMIT reported in the paper (PER 18.91 on validation set and 21.05 on test set) has clear gap compared to numbers reported in the literature after year of 2013. Basically, simple stacked bi-directional LSTM CTC recognizers should be able to achieve clearly lower validation/test set PER (on 39 Phones) than numbers reported in the paper.
I understand that achieving low PER on TIMIT (and low WER on LibriSpeech) is not the goal of this paper. However, the goal of NAS search is to search for competitive (or even state-of-the-art architectures) for using and future research. While the NAS-BENCH-ASR dataset is built upon TIMIT, it would be more convincing if stronger models with better validation/test PER are identified.

Regarding early stopping:
According to the last paragraph in page 4, it seems that the authors logged the validation loss/PER for each epoch, but only logged the test metrics at the end of the training. So, the test performance is not coming from the best epoch, this might have some effect to the validation/test performance correlation study.

Regarding transferability:
The authors claimed the transferability of the identified cell structure. The evidence is the correlation between Librispeech WER and TIMIT PER. To better support this claim, the authors may consider conduct study on some other corpus like SWITCHBOARD.

---

> ### Author Response · Authors · 2020-11-22
> **Addressing the questions**
>
> We would like to thank the reviewer for his/her time and effort in giving us valuable feedback. In the following we present our responses to his/her comments.
>
>
> **Re. 1. Why the design only uses unidirectional, rather than bidirectional LSTM? Is the paper focusing on on-device deployment?**
>
> Yes, it is. We will make it clear in the revised paper.
>
>
> **Re. 2. Regarding the performance on TIMIT.**
>
> Please see our reply to Reviewer-2.
>
>
> **Re. 3. According to the last paragraph in page 4, it seems that the authors logged the validation loss/PER for each epoch, but only logged the test metrics at the end of the training. So, the test performance is not coming from the best epoch, this might have some effect to the validation/test performance correlation study.**
>
> “final test PER” refers to the test PER of the best model according to its validation accuracy after 40 epochs of training - this is already mentioned in Section 4, third paragraph, second sentence. However, we understand that it’s easy to miss and we use the term even before it’s described in detail so we will make it clearer in the revised paper.
>
> **Re. 4. The authors claimed the transferability of the identified cell structure. The evidence is the correlation between Librispeech WER and TIMIT PER. To better support this claim, the authors may consider conduct study on some other corpus like SWITCHBOARD.**
>
> We agree that more use cases would naturally strengthen our claims but we’d argue that our current scope is already sufficient to make a case for NAS in the ASR domain (please see our reply to Reviewer 5’s comment 4 for more details). In summary: the results are self-contained and were obtained in a rigorous manner, they answer important questions regarding usage of NAS for ASR, and they required an already significant amount of engineering work and compute resources - similar to related work published in top conferences.

---

### Official Review · AnonReviewer1 · 2020-11-02

**Rating:** 7
**Confidence:** 3

**Review:**

The paper presents a study on neural architectural search on speech recognition. An new approach for NAS using convolutional models on speech is presented. Using the TIMIT dataset, ~8k different architectures are trained and evaluated. A study shows that the findings on TIMIT transfer well to a large-scale dataset, Librispeech.

Pros:
- The study is clearly novel, As far as I can tell this is the first NAS paper on speech.
- The finding on the transferability between TIMIT and Librispeech is significant.
- The paper is well motivated and well situated in the literature.

Cons:
- The databases selected makes the findings and the models limited to one domain, hence limiting the significance of the paper.

Detailed comments:
- My main concern with the paper is the choice of corpora: TIMIT and Librispeech are commonly used in ASR studies, but they are both composed on only clean, read speech in English. There is no way to know if the TIMIT results also transfer to another type of speech (conversational, acted, etc), to other recording conditions or to another language. Hence, the presented findings and models are only useful to ASR research focusing on this particular domain. I would encourage the authors to plan more studies with varied data as future work.

Overall, I think the paper is a very good first step towards more NAS-benchmark studies for speech, hence I vote for acceptance despite its limitation in terms of domain.

---

> ### Author Response · Authors · 2020-11-22
> **Transferability to other speech domain**
>
> We would like to thank the reviewer for his/her time and effort in giving us valuable feedback.
>
> We do agree that scepticism about transferability is justified but, as hinted by the reviewer, we also believe that it does not undermine our current results. Even if limited by the scope of our work, we think our contributions are significant in the context of NAS for ASR.

---

### Official Review · AnonReviewer5 · 2020-11-05
**NAS benchmark for ASR**

**Rating:** 5
**Confidence:** 5

**Review:**

This paper proposes a new experimental benchmark for ASR based on neural architecture search (NAS). NAS becomes one of the important machine learning/deep learning areas and has been widely studied in image classification and NLP tasks. This paper follows this trend and provides a NAS benchmark for ASR by using TIMIT. The paper also has a number of experiments by changing the neural network architectures and shows a lot of interesting findings. The paper is well written overall.

Strengths:
1) Providing a NAS platform for ASR
2) A lot of analysis in terms of the various architectural configurations and NAS algorithms
3) A discussion of applying such methods to the other database (librispeech).

Weaknesses:
1) The problem is too specific to ASR. It may not gain much attention from general machine learning researchers in ICLR
2) Not so much technical or algorithmic novelty, although I appreciate the authors' efforts for this new benchmark.
3) TIMIT is not a public/downloadable corpus, and its access is limited. I recommend the authors to try other easily accessible corpora (e.g., CMU an4 or "mini" Librispeech).
4) The analysis is mainly based on one corpus (TIMIT), and I'm not very sure about the finding and discussions are applicable to the other database.
5) No survey about the architecture search efforts in ASR. There is a lot of literature about NAS, evolution algorithm, a black-box search of ASR architectures.
6) The optimization hyper-parameters and input feature configurations should be considered as one of the search configurations. The architectures, input feature configurations, and optimization hyper-parameters are highly correlated.

Other comments
- Does the TIMIT experiment include a phone bi-gram model? This is a standard experimental setup.
- Figure 2 is an interesting finding. The paper says that it is different behavior compared with image classification benchmarks. Please discuss it by referring to the report about the image classification benchmarks.

---

> ### Author Response · Authors · 2020-11-22
> **Addressing weakness**
>
> We would like to thank the reviewer for his/her time and effort in giving us valuable feedback. In the following we present our responses to his/her comments.
>
>
> **Re. 1: The problem is too specific to ASR.**
>
> Although our work focuses mainly on ASR, we believe that our efforts in incorporating NAS in the ASR domain bridges an important gap in the literature and thus contributes to the both and related communities; and thus we believe our work will be of interest to many researchers. To further support our claim, we would like to kindly point out that all of the existing NAS benchmark works are also limited to only one task (e.g, image classification) and they have been successful in attracting the attention of many, e.g, NAS-Bench-201 was published as a spotlight at last year's ICLR.
>
>
> **Re. 3: TIMIT is not a public/downloadable corpus, and its access is limited.**
>
> We chose TIMIT mainly due to its popularity, small size, and high quality of phoneme-level transcriptions. In general, we agree that it’s a better practice to stick to the publicly available datasets, however, in this particular case there are a number of good reasons to use TIMIT:
> - Compared to CMU, TIMIT is more popular and therefore results contained in the dataset are easier to relate to the existing literature.
> - Compared to “mini” Librispeech TIMIT is a better choice in the context of our transferability experiments since it’s completely disjoint from the full Librispeech dataset.
> - It also gives us some insights into transferability of architecture search findings from a phoneme based system to a character/sub-word based system
> - Its transcriptions are hand verified, and balanced for phonetic and dialectal coverage.
> - Even though the TIMIT dataset is not open sourced, it is available to be used for non-commercial and academic purposes freely.
>
>
> **Re. 4: The analysis is mainly based on one corpus (TIMIT).**
>
> Similarly to our answer to the first issue raised by the reviewer, we’d like to point out that our approach is following existing NAS benchmarks and the choice to focus on a single corpus is inline with the existing literature. Even though we understand the reviewer's scepticism about transferability to other datasets, in the paper we only mention that - in the context of our search space, training methodology, etc. - correlation between TIMIT and Librispeech performance is high, which is an important finding potentially opening doors for more NAS research in the ASR domain. In general, we agree that “more is better” in ML research, but we’d argue that our choices regarding the scope of work are all well-justified considering the state of NAS and ASR research, and the presented findings are sufficient to be of importance to our chosen domain.
>
> **Re. 5: No survey about the architecture search efforts in ASR. There is a lot of literature about NAS, evolution algorithm, a black-box search of ASR architectures.**
>
> We admit that there is some NAS-related work for ASR which is currently missing - however, most of it seems to have been published very recently - after or around the time our manuscript was first submitted. We will update the revised paper and point to the relevant prior work (listed below).
>
> J. Kim et al. "Evolved Speech-Transformer: Applying Neural Architecture Search to End-to-End Automatic Speech Recognition." INTERSPEECH (2020).
>
> Y. Chen et al. "DARTS-ASR: Differentiable Architecture Search for Multilingual Speech Recognition and Adaptation." INTERSPEECH (2020).
>
> T. Mo et al. “Neural Architecture Search For Keyword Spotting.” INTERSPEECH (2020).
>
> L. He et al. “Learned Transferable Architectures Can Surpass Hand-Designed Architectures for Large Scale Speech Recognition.” arXiv preprint arXiv:2008.11589 (2020).
>
> A. Baruwa et al. “Leveraging end-to-end speech recognition with neural architecture search.” arXiv preprint arXiv:1912.05946 (2019).
>
> T. Veniat et al. “Stochastic adaptive neural architecture search for keyword spotting.” ICASSP (2019).
>
> H. Mazzawi et al. “Improving keyword spotting and language identification via neural architecture search at scale.” INTERSPEECH (2019).
>
>
> **Re. 6: The optimization hyper-parameters and input feature configurations should be considered as one of the search configurations.**
>
> Inclusion of additional hyper-parameters and input configurations in the search space would result in an exponential increase in configurations and would thus be computationally infeasible. To somewhat mitigate this, we decided to instead include results using two different sets of hyper-parameters. Please note that our decision to keep the search space limited to architecture only is aligned with the existing NAS benchmarks (as the problem of feasibility is common), and the decision to include results for two different sets of hyper-parameters is taking a step further to acknowledge the problem (since other benchmarks did not do that).

---

> ### Author Response · Authors · 2020-11-22
> **Other comments**
>
>
> **Re. 1: Does the TIMIT experiment include a phone bi-gram model?**
>
> The results presented in the paper are without a phone bi-gram language model (LM).
> We didn't use it in our experiments in order to avoid the confounding impact of LM and HPO for the LM, e.g., weighting factors. This helps to keep the architecture search for Acoustic Model (AM) tractable. Moreover, we did try including a LM for TIMIT in our initial experiments, but we observed the use of the bi-gram model trained with the TIMIT train set didn’t give a significant improvement in PER (i.e., only 0.2% absolute gain). We would add the details in the revised version of the paper.
>
>
> **Re. 2: Figure 2 is an interesting finding. The paper says that it is different behavior compared with image classification benchmarks. Please discuss it by referring to the report about the image classification benchmarks.**
>
> Will do in the revised version. Please see the table below which highlights the differences in terms of Spearman-r correlation:
>
>
>
> |  | NB1, CIFAR-10 | NB2, CIFAR-10 | NB2, CIFAR-100 | NB2, ImageNet-16 | NB-ASR, TIMIT (ours) |
> |---|---|---|---|---|---|
> | **Top 1000** |  0.356* | 0.855 | 0.613 | 0.827 | 0.210 |
> | **Overall** | 0.989 | 0.993 | 0.987 | 0.997 | 0.852 |
>
> \* note that NB1 contains much more models so the top 1000 models represent much lower percentage. From our observations this makes top 1000 correlation lower (i.e., if we took fewer models in other cases, to match percentages, correlation would also be lower for them as well) and is reflected by much higher discrepancy between Top 1000 and overall correlations compared to other cases

---

### Author Response · Authors · 2020-11-23
**Revised manuscript**

Our revised manuscript addresses comments raised by the reviewers as following:
- Extended related work to discuss NAS work in ASR.
- Clarification about early exit, use of regularizers/dropouts, design choise for unidirectional lstm, and no use LM model.
- Comparison of correlations between test and validation accuracies reported in the existingimage classification benchmarks (NB1 and NB2) and ours NB-ASR.

---

### Comment · ~Hyeoncheol_Jung1 · 2025-10-29
**Dataset link is broken**

Hello, I’m trying to reproduce your work and access the dataset,
but the dataset link (provided in the paper) seems broken.
Could you please update or share a new access link?

---

### Decision · Program_Chairs · 2021-01-07
**Final Decision**

**Decision:**

Accept (Poster)

**Comment:**

Good clarity: a NAS benchmark for ASR and results transferable across datasets. Although this is more specific to speech domain, building such a benchmark for speech is important for general NAS research, especially the papers finds different behaviors compared to image classification benchmarks.

The main factor for the decision is the clarity and importance for NAS in speech domain.